# Study on the Physical and Chemical Properties of Cement-Based Grout Containing Coal–Fly Ash

**DOI:** 10.3390/ma15248804

**Published:** 2022-12-09

**Authors:** Wanhong Guo, Shizhuo Zou, Shaochang Pu, Yu Zhou

**Affiliations:** 1Key Laboratory of Ministry for Efficient Mining and Safety of Metal Mines, University of Science and Technology Beijing, Beijing 100083, China; 2Sinohydro Foundation Engineering Co., Ltd., Tianjin 301700, China; 3PowerChina Road Bridge Group Co., Ltd., Beijing 100048, China

**Keywords:** fly ash, hydration reaction, heat of hydration, thixotropic ring, scanning electron microscopy

## Abstract

To study the physical and chemical properties of grout containing fly ash, Class II fly ash was used as a mineral admixture and mixed with silicate cement to produce grout, and the rheological properties, strength properties, hydration properties, and microscopic mechanism were studied. The results of the study showed the following. The incorporation of fly ash reduced the thixotropic area of the composite cement slurry, which facilitated pumping in the pipeline conveying process. The inclusion of fly ash reduced the yield stress and plastic viscosity of the cement paste, but the rheological index increased and then decreased with the increase in fly ash, and the composite paste had the lowest degree of shear thinning at 30% fly ash inclusion. The incorporation of fly ash reduced the hydration exothermic rate and total hydration exothermic amount of the composite slurry and prolonged the hydration induction period, but the promotion effect of fly ash on the hydration rate of cement was obvious at 10% fly ash admixture. The admixture of fly ash increased the empty volume of the composite slurry, but the effect on the most probable aperture was not significant, and the porosity of the system increased, resulting in a decrease in compressive strength. The effect of adding fly ash on the hydration products was reflected mainly by the C-S-H gel produced by cement hydration and the change in calcium alumina and Ca(OH)_2_. Fly ash does not directly participate in the hydration reaction of cement, but it can promote cement hydration and increase the reaction rate of cement. By analyzing the rheological properties, mechanical properties, and hydration properties of fly ash composite cement paste, the comprehensive analysis found that the rheological properties are excellent when the fly ash admixture is 20–30%, and the water–cement ratio can be reduced to improve the strength without affecting the pumping demand.

## 1. Introduction

A mineral admixture is a fine, inorganic mineral powder that can improve the performance of fresh grout and hardened grout when added in the preparation of grout. The addition of mineral admixtures to grout not only reduces the cement content—thus reducing the heat of hydration and the cost of production—but also enhances the durability of the grout [1]. In the context of China’s development of a dual carbon strategy, the use of slag powder is an important method to achieve the low carbonization of cement and grout [2].

Usually, the amount of mineral admixture is greater than 5% of the cement dosage, and the fineness is the same as or finer than the cement fineness. The mineral admixtures are fly ash [3], granulated blast furnace slag, silica fume, limestone powder, steel slag powder, phosphate slag powder, zeolite powder, and composite mineral admixtures [4]. Fly ash is a fine powder that is a byproduct of the burning of pulverized coal in electric generation power plants, which can increase the fluidity of the cement slurry in fresh cement slurry and also effectively control the slump loss of grout [5,6,7]. Fly-ash-containing grout has a dense structure as well as excellent interfacial bonding properties and exhibits good physical and mechanical properties [8]. Due to its low cost, it is now widely used to improve grout performance and reduce costs. Herath et al. [9] mentioned that mineral admixtures can replace 20% to 35% of cement in equal amounts to formulate grout under the premise of improving the performance of grout. Vasumithran et al. [10] conducted indoor tests based on cement-based grout composed of silica fume, fly ash, and fine sand, and they found that the particle size of the constituent materials was one of the most important influencing factors for the viscosity of the grout samples and that the grout with the addition of mineral admixtures was more viscous. Ma et al. [11] conducted a study on the shear-thickening behavior of fly ash (FA), slag (SL), and limestone powder (LP) instead of cement slurry and explored the effect of the materials on the rheological curve parameters and shear-thickening effect. Zhang et al. [12] comprehensively investigated the effect of the mineral admixture type and dosage on the rheological and mechanical behavior of a newly developed backfill material, cementitious foam backfill (CFB), with the selected variable materials being fly ash (FA), ground granulated blast furnace slag (GGBS), and quicklime. It can be found that the use of fly ash as a mineral admixture not only improves the grout performance substantially but also serves as an important component of backfill material cementitious foam [12], natural hydraulic lime-based mortar [13], clay–cement composite grout [14], etc. Scholars have often measured the comprehensive performance of grout containing mineral dopants based on the study of both the macro-mechanical properties [15,16] and micro-fine characteristics [17,18] of the composite materials and have made progress in stages.

However, there are still major shortcomings in domestic and international research; the research conclusions are simple and general, and the research mechanism of fly ash-containing grout is not sufficiently understood, which is not conducive to the promotion and use of fly-ash-containing grout. Therefore, this paper uses Class II fly ash produced by Henan Hengxiang New Materials Co., Ltd. (Zhengzhou, China) as a mineral admixture, mixed with silicate cement to produce grout, and studies the material physical and chemical properties of fly-ash-containing grout in terms of four aspects: rheological properties, strength properties, hydration properties, and micro-mechanics. The results of the study can provide guidance for the grouting and impermeability of the Longtan coal mine transportation roadway project.

## 2. Materials and Methods

The engineering background of this study relies on the Longtan coal mine in Guang’an, Sichuan Province, which has geological reserves of 84.95 million tons and a coal production scale of 1.5 million tons/year. It is the main source of coal for the city’s thermal power plant. The Longtan coal mine transportation roadway is a control project aiming to guarantee the normal production of the Longtan coal mine. It crosses the lower part of the reservoir bed of the Piziguan River at Longtan Reservoir from west to east. The main flat cavern and its interior are partially lined with concrete and partially supported by anchor rods, and most of the cavern is a bare tunnel, as shown in Figure 1. At present, there are several water-gushing points in the transportation lane. To guarantee the normal operation of the transport lane under the water storage condition in the riverbed, it is proposed to grout the whole area of the transport lane for anti-seepage treatment at a later stage. Therefore, it is of great research significance and economic value to study the physical and chemical properties of grout, to select grout with a reasonable proportion, and to reduce the cement dosage. The project intends to use Class II fly ash as a mineral admixture, mixed with silicate cement to produce grout, to perform a quantitative study of the effect of different amounts of fly ash on the grout effect, the specific raw materials, and their combination, as shown below.

### 2.1. Raw Materials

#### 2.1.1. Cement

P.O. 42.5 ordinary silicate cement produced by China United Cement Group Co., Ltd. (Beijing, China) was used for the test, mainly consisting of silicate cement clinker tricalcium silicate (C_3_S), dicalcium silicate (C_2_S), tricalcium aluminate (C_3_A), and tetracalcium iron aluminate (C_4_AF). Figure 2a shows the microscopic morphology of the ordinary silicate cement used in this experiment, and the cement particles are irregular particles with sharp angles. The physical and mechanical properties of the cement are shown in Table 1, and the chemical composition is shown in Table 2.

#### 2.1.2. Fly Ash

The fly ash used in the test was the Class II fly ash produced by Henan Hengxiang New Materials Co., Ltd. Figure 2b shows the microscopic morphology of the Class II fly ash used in the test under backscattering electron microscopy (BSE) at 300 magnification. The chemical composition of the fly ash samples is shown in Table 3, and the particle size distribution is shown in Figure 3.

#### 2.1.3. Water

The test process used tap water from the laboratory to mix the cementitious materials.

### 2.2. Test Ratio

The test net slurry mixes are shown in Table 4, with a water to glue ratio of 0.45 (glue is a cement/fly ash mixture) and fly ash admixtures of 0, 10%, 20%, 30%, and 40% in the samples denoted as C, F10, F20, F30, and F40, respectively.

### 2.3. Test Methods

#### 2.3.1. Rheological Test

The rheological curves of the freshly mixed slurry after 5 min, 60 min, and 120 min of mixing were measured using a rheometer. They were all sealed before slurry testing and kept in the standard maintenance box. Mixing was performed at the same time and at the same speed before testing at different time timepoints. The rotor of the rheometer was placed at a distance of 2 cm from the bottom, the slurry height over the rotor was 2 cm, the rotor was 4 cm long, and the rotor was located in the middle of the slurry.

Flow curves: To eliminate the possible effect of particle agglomeration, the samples were pre-sheared for 30 s and then left to stand for 10 s. After this, the shear rate was increased linearly from 0 to 100 s^−1^ in 100 s and then decreased linearly from 100 s^−1^ to 0 in 100 s. The test protocol for the flow profile is shown in Figure 4. The lower shear curve was used to calculate the rheological parameters. It can be seen that the cement slurry exhibited shear-thinning, Bingham, and shear-thickening behaviors at low, medium, and high shear rates, respectively. In this study, most of the cement slurries showed a linear decrease and then a nonlinear change at low shear rates as the shear rate was decreased. To obtain comparable results, the HB model was used to fit the rheological curves.
(1)$$\tau =\tau_{0}+k{\dot{\gamma }}^{n}$$

Here, *n* denotes the rheological index, and when its value is lower (higher) than 1, it indicates shear thinning (thickening); *τ*_0_ denotes the yield stress (Pa); *k* denotes the fluid consistency coefficient, and the larger the value of *k*, the more viscous the fluid—that is, the greater the resistance to fluid flow.

Using the least squares method, the empirical equation that can be used to calculate the equivalent plastic viscosity was derived as follows:(2)$$\mu =\frac{3k}{n+2}{\dot{\gamma }}_{max}^{n-1}$$

#### 2.3.2. Strength Test

The specimen size of 160 mm × 40 mm × 40 mm was used for the test research, and the complete preparation process of the specimen was as follows: (1) the standard mold was cleaned and dried and then brushed with oil to ensure the integrity of the specimen when it was demolded; (2) the material was weighed according to each design ratio; (3) the cement and fly ash were mixed and poured into the mixer bin after the initial stirring, and tap water was poured for 4 min for full mixing; (4) the fully mixed mortar was poured into the mold and shaken for 3 min to fully expel air, to reduce the influence of air bubbles on the strength property test; (5) the specimens in the molds were scraped flat for initial setting, and demolded after 1 d. The demolded specimens were placed into a 20 ± 2 °C constant-temperature maintenance box for four time periods of 1 d, 3 d, 7 d, and 28 d. Uniaxial compressive and uniaxial flexural strength tests were conducted on the corresponding specimens when the corresponding ages were reached. To reduce the dispersion of the test data, three parallel specimens of each ratio and age were prepared. Specimen production, maintenance, and testing processes followed the national standard GB/T 17671-1999 “Cement Sand Strength Test Method (ISO Method)”. The cement–fly ash net slurry mixes used for the tests are shown in Table 4. The test machine was a microcomputer-controlled electro-hydraulic servo pressure tester with a force-controlled loading method and a loading rate of 100 N/s. The test procedure is shown in Figure 5.

#### 2.3.3. Micro-Mechanism Study

To better study the microscopic mechanism of the hydration reaction of cementitious grout containing fly ash, the microscopic morphological characterization of grout at different ages was carried out. The specimens were cut into blocks by intercepting the middle section from the pre-prepared specimens of different ages, immersed in anhydrous ethanol for 24 h to stop the hydration of the grout, and then dried in an oven at 40 °C for 30 min, impregnated with low-viscosity resin, and polished using an automatic sample grinder. The drying and impregnation processes did not cause any visible cracks. A Hitachi TM-4000 scanning electron microscope was used to study the microscopic properties of the specimens. At present, the backscattered electron microscopy (BSE) technique has been widely used in the micro-mechanical study of cementitious materials, and this technique can better overcome the defect of SEM wherein it is difficult to clearly observe the morphology in the microscopic morphology of the hardened slurry of the mineral admixture (fly ash in this experimental study), and the application of this technique can also allow us to observe the degree of hydration of the cement and mineral admixture in the hardened slurry.

## 3. Results

### 3.1. Rheological Performance Study

Rheology is the study of the flow of matter. Real objects (or materials) will be deformed (or flow) under the action of external forces; according to its nature, deformation can be divided into elastic deformation, viscous flow, and plastic flow. Cement slurry exhibits different rheological behaviors with changes in the type of mineral admixture and additives, the amount of admixture, and the ambient temperature. The study of the rheological properties of cement slurry with different ultrafine mineral admixtures can provide a basis for optimizing the ratio of cementitious materials and adjusting the preparation process of admixtures, thus improving the construction performance of grout and enhancing the application effect. The study of the rheological properties of cement slurry containing ultrafine mineral admixtures is important to ensure the later strength while maintaining the high working performance of the cement slurry.

#### 3.1.1. Thixotropic Ring

Figure 6 shows the thixotropic ring curve of the composite cement slurry containing fly ash. When the shear rate increases continuously from 0 to a constant value, and then gradually decreases from this constant value to 0, the stress is measured with the change in shear rate. When the shear rate increases, the shear stress–shear rate curve is a rising curve; the shear stress–shear rate curve becomes a falling curve when the shear rate decreases. From Figure 6a, it can be seen that the rising curve of the pure cement slurry and fly-ash-containing composite cement slurry is above the falling curve. The area enclosed by the upward and downward curves is called the thixotropic area, which indicates the energy dissipation during the shearing of the slurry. The larger the thixotropic ring area, the more energy is required to restore the slurry to its original state [19]. The addition of a small amount of fly ash increased the thixotropic ring area of the composite slurry, and as we continued to increase the amount of fly ash, the thixotropic ring area of the composite slurry began to decrease, which was more obvious at 40% of the mixture. When the cement slurry undergoes the shear process, flocculation, and de-flocculation at the same time, at a high shear rate, the flocculation structure is destroyed, and it is restored when the shear is stopped or the shear rate is reduced. The particle size of fly ash is smaller than that of cement, and when the amount of fly ash is small, the fly ash can fill in the spaces between cement particles, which can lead to the release of the clusters between cement particles with deflocculation; this increases the contact area between the cement particles and water. At the same time, fly ash acts as a nucleating matrix for C-S-H hydration, which reduces the nucleation barrier and promotes cement hydration to a certain extent. Moreover, the flocculation structure of the composite slurry with a larger amount of fly ash admixture is significantly lower than that of the pure cement slurry, and the admixture of fly ash reduces the amount of the flocculation structure in the composite slurry, which is related to the amount of cement clinker, because fly ash does not participate in the hydration reaction in the early stage of hydration. In addition, the shear stress of the composite slurry containing 10% fly ash is not significantly different from that of pure cement, or it is even slightly higher than that of pure cement, which is because the admixture of a small amount of fly ash increases the hydration rate of cement, which causes the flocculent structure created by the hydration products in the composite cement slurry to increase, and it also causes the shear stress of the slurry to increase. The increase in the cement clinker reaction rate in the composite slurry could not compensate for the decrease in hydration products caused by the decrease in cement dosage after the fly ash dosage exceeded 10%, which led to a decrease in the flocculent structure in the composite slurry, leading to a decrease in shear stress, and the higher the fly ash dosage, the more obvious the decrease in shear stress. The effect of fly ash on the apparent viscosity of the cement slurry is shown in Figure 6b. From Figure 6b, it can be seen that the fly ash admixture has a great influence on the rheological curve of the fresh slurry, and the admixture of fly ash reduces the apparent viscosity of the cement slurry. The higher the admixture, the smaller the viscosity, which is related to the particle size distribution and the particle morphology of fly ash. The particle size distribution of cementitious materials is an important factor affecting the shear stress and viscosity. The number of tiny fly ash particles is much higher than that of pure cement, while the number of larger particles is the same as or smaller than that of pure cement particles. Smaller fly ash particles can fill the voids, and the filling density of the cement slurry increases to release pore water, which facilitates the increase in water film thickness. However, the water demand of fly ash is small; regardless of the amount of fly ash, the apparent viscosity of the cement slurry is reduced, indicating that the filling effect of fly ash has a greater influence on the rheology, and the high water demand effect of cement particles is dominant, so the plastic viscosity increases with the increase in fly ash admixture. The study shows that the apparent viscosity can be used to roughly compare the advantages and disadvantages of the flowability of the composite slurry [20], indicating that fly ash is beneficial to the flowability of the cement slurry, and the flowability of the slurry increases with the increase in the fly ash admixture.

As shown in Figure 6c, the shear stress of the rheological curve of the composite cement slurry increased after 60 min of hydration, and the higher the amount of fly ash admixture, the smaller the increment in shear stress. It is worth noting that the shear stress difference between the cement composite slurry with 10% fly ash admixture and pure cement was significantly increased compared to that at 5 min. This indicates that the cement is the main body of the hydration reaction in the early stage, and the admixture of fly ash reduces the amount of cement clinker. The hydration products are reduced, while fly ash only plays a general role in promoting the hydration of the cement clinker, which starts to weaken after one hour. The thixotropic ring area of the cement slurry increased significantly compared with that at 5 min. This increment in thixotropic ring area was more obvious at a low admixture level, while the difference in the thixotropic ring area was not significant at a high admixture level. This is due to the larger cement clinker substitution with 40% admixture and the smaller increment in hydration products. The plastic viscosity of the composite cement slurry also increased compared to 5 min, as seen in Figure 6d, which is also related to the hydration of the cement. The change in the plastic viscosity of the composite cement slurry at 60 min is the same as that at 5 min.

Figure 6e shows the shear stress–strain curve of the composite cement slurry at 120 min. From Figure 6e, it can be seen that the shear stress in the rheological curve of the composite cement slurry increases after 120 min of hydration, and the higher the admixture of fly ash, the smaller the increment in shear stress. The thixotropic ring area of the cement slurry increased compared with that at 60 min, and the increment in thixotropic ring area was still more obvious at a low admixture level. The increment in the thixotropic ring area of the composite cement slurry remained insignificant with 40% admixture. This is because the cement clinker substitution is larger with 40% admixture and the increment in hydration products is smaller. Figure 6f shows the apparent viscosity change curve of the composite cement slurry at 120 min, and the apparent viscosity of the composite cement slurry increases at 120 min compared to 60 min. This is because of the hydration of cement; fly ash does not participate in the reaction in the early stage of hydration and only affects the reaction rate of cement particles.

Combining the thixotropic ring curves of the composite cement paste containing fly ash at different times, it can be found that the thixotropic ring curves for different levels of fly ash content change in an essentially consistent pattern with increasing time. Shear stress and plastic viscosity decrease with increasing fly ash admixture due to the fact that the increase in fly ash fraction reduces the proportion of cement clinker, which causes a reduction in hydration products and also weakens the effect of the increasing hydration reaction time on the increment in hydration products, so that the performance of the cement paste with a large amount of fly ash admixture is less affected when the time is varied. At 5 min, the shear stress of the cement slurry with 10% fly ash admixture is not significantly different from that of the pure cement slurry. Fly ash has an enhancement effect on the cement hydration process, so, with a smaller amount of admixture, the effect brought about by the replacement of the cement clinker components can be weakened, but the enhancement effect will be gradually weakened as the hydration reaction proceeds.

#### 3.1.2. Rheological Parameters

Considering the effect of time, the rheological properties of the slurry of composite cementitious materials containing different fly ash admixtures at different times are studied in this section, and the rising shape of the selected thixotropic ring curve constitutes the rheological curve of shear rate versus shear stress, as shown in Figure 7. From Figure 7, it can be seen that the fly ash admixture and hydration time have a great influence on the rheological curve of the freshly mixed slurry. The shear stress of the composite cementitious material slurry containing fly ash increases gradually with time, which is related to the hydration of the composite cementitious material. The admixture of fly ash is an important factor affecting the rate of increase in shear stress. The admixture of fly ash reduces the shear stress of the slurry, and the larger the amount of admixture, the smaller the shear stress of the cement slurry.

Table 5 shows the rheological parameters obtained by fitting the slurry rheological curves of composite cementitious materials containing fly ash at the water to glue ratio of 0.5. As shown in Table 5, the yield stress gradually decreases with the increase in common fly ash admixture. This change is mainly related to the water film thickness, which is the main factor affecting the rheological properties of the freshly mixed slurry of cementitious materials [21], while the particle-filling effect releases pore water with the increase in specific surface area. The water demand has a competitive effect on the water film thickness, and the change in water film thickness depends on which one is the dominant factor. When fly ash is mixed, the water film thickness increases, and the internal friction between particles decreases. This is mainly because the fly ash particle size is smaller than the cement particles, which indicates that fly ash can cause the filling density to increase to release the pore water. Although the total specific surface area of the system increases with the incorporation of fly ash, the water requirement of fly ash is low, which means that the impact of the increase in specific surface area on the reduction in water film thickness is smaller when fly ash is incorporated. The incorporation of fly ash increased the amount of free water, decreased the amount of pore water, and increased the thickness of the water film, which was beneficial to the rheological properties of the composite slurry. It is stated that at the same moisture content, this will increase the amount of excess water that forms a water film covering the solid particles to improve the flow, or allow a lower ratio of water to cementitious material to improve the strength at the same flow requirements. The spherical fly ash particles act as “spherical teeth” in the cement slurry to overcome the internal friction between the fly ash and the cement, while reducing the agglomeration of the flocs and releasing the water from the flocculation structure. The surface activity of fly ash is low and the water requirement is relatively low, which enhances the water-reducing effect, while the “ball bearing” of fly ash microbeads also plays a certain role in the water reduction [22], so fly ash is used as a mineral admixture with excellent performance to improve the fluidity of ultra-high-performance grout in various engineering applications. Therefore, as the amount of ultrafine fly ash increases, the yield stress decreases rapidly. Ordinary fly ash particle sizes are smaller than those of cement particles, and they can fill in the cement particle gaps to release pore water, but this increases the area of mutual contact between particles to a certain extent, increasing the interaction force between particles; with the increase in the amount of admixture, the particle stacking density continues to increase. However, because the fly ash has spherical particles, it results in inter-particle motion dominated by rolling friction, which greatly reduces the inter-particle internal friction, manifesting as a decrease in plastic viscosity. In addition, the water film thickness also plays an important role in the plastic viscosity. The yield stress and plastic viscosity of the composite cementitious material slurry containing common fly ash gradually increased with time, and the increase was more obvious for the specimens with less fly ash. The slurry forms a network structure under the action of inter-particle gravitational force and hydration, and fly ash can effectively increase the hydration rate and generate more network structures in the same time.

Figure 8 shows the rheological index of composite cement slurry, and the rheological index is related to the degree of shear thinning. When the rheological index *n* is 1, the HB model becomes the Bingham model, which means that the rheological curve of the slurry is linear; when *n* is greater than 1, the fluid is swelling fluid, and the apparent viscosity of the fluid increases with the increase in shear stress; when *n* is less than 1, the fluid is pseudoplastic fluid, and the apparent viscosity of the fluid gradually decreases with the increase in shear rate. The cement slurry before and after the addition of fly ash showed obvious pseudoplasticity characteristics, where the pure cement slurry was more pseudoplastic than when fly ash was added. As can be seen from Figure 8, the rheological index of the fresh slurry with the increase in fly ash admixture shows a change law of increasing and then decreasing, and all the rheological index values are above 0.72. The larger the rheological index, the lower the degree of shear thinning. At present, scholars focusing on rheological shear thickening and shear thinning mainly explain these phenomena via cluster theory and ordered disorder theory [23,24]. Cluster theory points out that the more the slurry is destroyed in the shear process, the greater the degree of change in the network structure, and the greater the degree of shear thinning. Meanwhile, ordered disorder theory points out the state of inter-particle motion in the slurry: when the inter-particle motion is orderly, the slurry shows shear thinning; when the inter-particle motion is disorderly, the slurry shows shear thickening. The variation law of the shear thinning degree of the composite cement slurry containing fly ash is the result of the joint action of the two theories. The admixture of fly ash reduces the amount of hydration products in the composite slurry, resulting in a reduction in the cluster structure, and the degree of network structure change gradually decreases with the increase in fly ash admixture. In addition, the particle spacing in the slurry decreases when fly ash is incorporated, and the probability of inter-particle collision increases. The state of unidirectional flow of particles between layers is easily broken by particles in adjacent laminar flow, and the degree of shear thinning of the slurry becomes smaller. Continuing to increase the amount of fly ash, the degree of shear thinning of the composite cement slurry increases. This indicates that the rheological index is mainly related to the inter-particle motion state at high yield with larger cement substitution, and the orderliness of inter-particle motion increases with a large amount of admixture.

As the hydration reaction proceeds, the yield stress and plastic viscosity increase, but when the percentage of fly ash increases, the yield stress and plastic viscosity decrease. The particle size of fly ash particles is smaller than that of cement particles, so they will fill in the gaps between cement particles, which will not only reduce the pore water, leading to an increase in water film thickness, but also act as a bearing to reduce the internal friction of cement. The rheological index of the composite slurry shows a trend of increasing and then decreasing with the increase in fly ash admixture. The addition of fly ash breaks the orderliness of the slurry, and the decrease in hydration products also leads to a decrease in the graph cluster structure. However, when the amount of fly ash is large enough, the orderliness of the cement slurry system will gradually increase.

### 3.2. Strength Characteristics Study

#### 3.2.1. Compressive Strength

Figure 9 shows the uniaxial compressive strength of the fly-ash-containing cement-based grout at four ages of 1 d, 3 d, 7 d, and 28 d. The compressive strength of pure cement grout increased from 6.50 MPa at 1 d to 12.34 MPa, 15.77 MPa, and 21.58 MPa, with growth rates of 89.8%, 142.6%, and 232%, respectively, which shows that the initial hydration strength of pure cement grout is low and it has a substantial increase in compressive strength as the degree of hydration reaction deepens. The 28 d compressive strengths of specimens F10, F20, F30, and F40 increased by 268%, 255%, 374%, and 451%, respectively, compared with the 1d strength, which indicates that the early hydration of cementitious grout with fly ash admixture is slower than that of pure cementitious grout, and with the advancement of the curing age, the cementitious grout with different fly ash admixture has greater strength than pure cementitious grout. This indicates that the activity of fly ash is fully stimulated in the later hydration reaction, and the admixture of fly ash is beneficial to the strength increase in cementitious grout. It can be seen from the figure that the compressive strength of pure cement grout is higher than that of cement-based grout containing fly ash in the four age tests from 1 to 28 d. As an active mineral admixture, the increase in the amount of fly ash decreases the amount of cement clinker in the grout, leading to a decrease in the hydration products, and thus the compressive strength in the overall age is lower than that of pure cement grout, and it decreases further with the increase in the amount of admixture.

#### 3.2.2. Flexural Strength

Figure 10 shows the uniaxial flexural strength of the fly-ash-containing cement-based grout at four ages of 1 d, 3 d, 7 d, and 28 d. From Figure 10a, it can be seen that the flexural strength of pure cement grout (C) has a clear advantage in the initial flexural strength of hydration at 1 d. The strength of specimen F10 is only slightly (10.2%) lower than that of pure cement grout, and the flexural strength of cement-based grout gradually decreases with the increase in fly ash admixture. The higher the fly ash admixture, the lower the flexural strength. Since fly ash does not participate in the hydration reaction in the early stage of hydration, it mainly plays the role of dense filling in the grouting system, and the hydration products are less than those in pure cement grout, which leads to a reduction in flexural strength; the flexural strength also decreases further with the increase in fly ash admixture. In the flexural strength tests at four ages from 1 to 28 d, the pure cement grout had the advantage of greater flexural strength performance. The 28 d uniaxial flexural strength of cementitious grout specimens F10, F20, F30, and F40 with fly ash admixture increased by 76.4%, 137%, 151%, and 206%, respectively, compared to the 1 d flexural strength. The flexural strength of cementitious grout specimens was mainly determined by the interfacial bond strength of the material, while the admixture of fly ash seemed to have a negative effect on the early interfacial bond of the grout. The early strength of the cementitious grout specimens with fly ash admixture was lower, but the later flexural strength of the specimens increased with the extension of the curing age, and the later hydration properties of fly ash at an age greater than 28 d need to be further investigated in subsequent tests.

The admixture of fly ash has a negative effect on the strength of cementitious grout, and the greater the admixture, the greater the effect, mainly because the admixture of fly ash reduces the proportion of cement clinker, which in turn significantly reduces the hydration products and causes a reduction in strength. With the extension of the curing time, the strength of the specimens with a larger percentage of fly ash grows faster, indicating that the activity of fly ash will be fully excited in the late stage of the hydration reaction.

### 3.3. Study of Hydration Properties and Micromechanisms

#### 3.3.1. Heat of Hydration

Generally speaking, the hydration process of cement can be divided into three stages: in the first stage, cement particles and water undergo contact and reaction, and the exothermic rate is very fast, but, due to the presence of gypsum, a layer of passivation mold will be formed on the surfaces of cement particles, so that the exothermic rate is reduced; in the second stage, the cement hydration heat release rate is the fastest, and the cement particles also grow very quickly; in the third stage, the hydration products of cement are gradually thickened on the surfaces of cement particles, and the hydration exothermic rate of cement is gradually reduced; the reaction at this time is controlled by diffusion [25]. Figure 11a shows the exothermic rate of hydration of the composite cement slurry, which can be roughly divided into five stages: pre-induction (15 min), induction (2–4 h), acceleration (4–8 h), deceleration (12–24 h), and stabilization. The first exothermic peak of hydration occurs in the pre-induction period, and the main causes are the production of calcium alumina and the heat of dissolution of C_3_S. As shown in Figure 11a, in the pre-induction period, the first exothermic peak is the largest for pure cement, followed by that for the composite slurry containing 10% fly ash, and it is the smallest for the composite slurry containing 40% fly ash. This is consistent with the variation law of yield stress and plastic viscosity with time, so the hydration reaction is the most important factor affecting the rheology of cement slurry through time. Entering the hydration induction period, the surface of C_3_S is covered by a relatively high calcium and silicon hydration product cladding layer, which prevents the further hydration of the cement. Fly ash delays the hydration induction period of the cement slurry, which is mainly related to the pre-induction period, where the cement hydration rate is faster and reaches the thickness of the cladding layer earlier. Entering the accelerated hydration period, the hydration products of the envelope layer are transformed into the more permeable Class Ⅱ hydration products due to the Class II phase change. C_3_S rapidly hydrates to form C-S-H gel and CH, reaching the second exothermic peak of hydration. In addition, the nucleation effect of fly ash and the effect of increasing the effective water to glue ratio both accelerate the hydration of cement and generate more hydration products. After the second exothermic peak of hydration, hydration enters the deceleration period and then enters the stabilization period. After entering the stabilization period, the hydration rate is extremely low.

Figure 11b shows the total exothermic heat of hydration of the composite cement slurry. As shown in Figure 11b, the total exothermic hydration of pure cement is the largest, with a larger growth rate of the curve; the total exothermic hydration of composite cement containing 10% fly ash is the second largest, while the total exothermic hydration of 40% fly ash composite cement is the smallest. This is consistent with the first exothermic peak of hydration and the variation law of rheological parameters with time. The amount of exothermic heat of hydration can indirectly represent the amount of flocculation structure generated in the cement slurry, which causes the rheological properties of the cement slurry to change. Between 2 h and 20 h, the rate of increase in the total exothermic heat of hydration of the composite cement slurry increases further, but the total exothermic heat of hydration is still lower than that of the pure cement, and the difference gradually increases. After 20 h, the growth rate of the exothermic heat of hydration curve gradually tends to level off, and the change pattern is the same as that at 2–20 h. However, the difference between the total amount of exotherm of different mineral admixtures and that of pure cement continues to increase.

Compared with pure cement, the size of the exothermic peak and total exothermic volume decreases with the increase in fly ash admixture when fly ash is mixed. Moreover, due to the presence of fly ash, the time to reach the second exothermic peak is also delayed compared to pure cement, but the amount of hydration products in the later stages of hydration increases due to the nucleation effect and the effect of increasing the effective water–cement ratio.

#### 3.3.2. Microscopic Morphology

The microscopic morphology of the fresh fracture of each specimen of hardened slurry samples at different hydration ages was observed in high vacuum mode using scanning electron microscopy (SEM). The hydration products as well as the microstructure can be roughly determined from the microscopic morphology of the hydration products in Table 6. The microscopic morphology of the cement–fly ash composite cementitious material-hardened slurry at 3 d of age under standard curing conditions is shown in Table 7. As can be seen from Table 7, a large number of needle–rod calcium alumina—Ca(OH)_2_—crystals can be observed in the pure cement specimen, with a smaller number of C-S-H gels generated and a looser overall structure. A large number of hydration products were generated in specimen F10, but the overall microstructure was looser than that of pure cement. Since only 10% of fly ash was present in specimen F10, it was difficult to see fly ash particles at low magnification. Fly ash particles can be observed in Table 7, with C-S-H gels and calcarenite around the unhydrated fly ash particles, and these hydration products grow toward the interior of the pores. The bonding between the hydration products is not tight and pores can be clearly observed. Spherical fly ash particles were clearly observed in specimens F30 and F40. In addition, flaky Ca(OH)_2_, needle–rod caliche, and C-S-H gels were also observed. The fly ash particles were covered with hydration products around the fly ash particles, but the surfaces of fly ash particles were very smooth, which indicated a very low degree of reaction of fly ash at the age of 3 d of hydration, which was consistent with the results of the heat of hydration of fly ash. Compared to specimen F20, the amount of hydration products in specimens F30 and F40 was lower and the degree of denseness was the worst. The reason for this is that a large amount of fine cement clinker is replaced and the hydration products are reduced, thus increasing the density of the hardened slurry to be reduced. In addition, the large fly ash particles are poorly bonded to the surrounding hydration products and become a weak point of force in the hardened slurry. The bonding between the hydration products is not tight and pores can be clearly observed.

Table 8 shows the microscopic morphology of the cement–fly ash composite cementitious material-hardened slurry at 7 d of age under standard curing conditions. From Table 8, it can be seen that the cement particles with a smaller particle size in the pure cement-hardened slurry have been further hydrated, and the particles are surrounded by a thick layer of hydration products. Meanwhile, the cement particles with a larger particle size have a lower degree of hydration and relatively fewer hydration products around them. At 7 d of hydration, the structure of the pure cement-hardened slurry was significantly more dense at 3 d of age. The hydration products in specimen F10 are greater and the density is comparable to that of pure cement. This is due to the microaggregate effect and the dilution effect of fly ash, which promotes the hydration of cement. Unhydrated cement particles with a larger particle size, and fly ash particles, can be observed in specimen F20. The smooth surface of fly ash particles indicates that no reaction of fly ash particles occurred at 7 d of hydration and they only played a filling role in the hardened slurry, which was more obvious in specimens F30 and F40. As can be seen from Table 8, many unhydrated fly ash particles could be observed in specimens F30 and F40. The smaller fly ash particles can be used as a microaggregate filling phase, and the larger fly ash particles are not tightly connected to the surrounding hydration products and are a weak point of force in the hardened slurry. This is one of the reasons for the lower early compressive strength of the specimens after the incorporation of fly ash.

The larger the calcium–silica ratio of the grout, the more susceptible the grout is to sulfate attack and alkali–aggregate reaction [26]. In addition, alumina will also decompose and recrystallize, resulting in poor stability. Generally, one of the purposes of mixing volcanic ash materials and silica fume with these admixtures is to improve the calcium–silica ratio of grout, so that the mechanical properties and durability of grout can be improved. The theoretical calcium silicate ratio of hydrated calcium silicate is between 1.7 and 2.3. Above this range, the grout will experience problems. Reducing the calcium–silica ratio of C-S-H gel in cementite is one of the methods to improve the mechanical properties and durability of grout. Calcium silicate hydrate (C-S-H) is one of the most important hydration products of cement. The current study shows that C-S-H gel accounts for approximately half of the volume of cement slurry, and C-S-H has a great influence on the macroscopic properties (such as strength, shrinkage, creep, etc.) of cement-hardened slurry as well as grout. Figure 12 shows the calcium–silica ratio of the hardened slurry at the age of 3 d, and it can be seen from the figure that the calcium–silica ratio of the C-S-H gel in the hardened slurry gradually decreases with the increase in the fly ash admixture. This indicates that in the early stage of hydration, the anti-sulfate erosion performance of the specimen is inversely proportional to the amount of fly ash admixture.

#### 3.3.3. Hole Structure

The pore structure is an important part of the grout microstructure and has an important influence on the macroscopic properties, such as strength and permeability, of the grout. Aggregates in grout are wrapped by hardened grout, and the pore structure properties of hardened grout have a greater impact on the performance of the grout. The structure of the pores is more important than the porosity regarding the macroscopic properties of the grout. Wu [27] classified pores into four classes: harmless pores (<20 nm), less harmful pores (20–50 nm), harmful pores (50–200 nm), and multi-harmful pores (>200 nm). The study of the pore structure of hardened slurry can help us to understand the effect of ultrafine mineral admixtures on the microstructure of hardened slurry in depth.

Figure 13 shows the pore structure of the hardened slurry of composite cementitious material containing common fly ash at the water to glue ratio of 0.45, at the age of 28 d, under standard curing conditions. The incoming mercury differential curve of the composite cementitious material-hardened slurry can reflect the pore size distribution characteristics of the hardened slurry. From Figure 13a, it can be seen that the most probable aperture of the hardened slurry is almost unchanged with the admixture of fly ash, but the pore volume increases. This indicates that the admixture of common fly ash increases the porosity of the hardened slurry, and the void volume continues to increase with the increase in admixture. Fly ash does not participate in the hydration reaction at the early stage of hydration, but only plays the role of filling and compacting. The pure cement sample has the smallest pore volume, the largest bulk density, and the smallest porosity. It can be seen from Figure 13b that the admixture of fly ash increased the volume of pores with a size of less than 100 nm, and the greater the admixture of common fly ash, the greater the increase. A greater number of large pores in the hardened slurry will adversely affect the strength, so the strength of composite cementitious material mastic sand mixed with a small amount of common fly ash at the age of 360 d of hydration, as described in the literature, is higher than the strength of pure cement mastic sand. The admixture of fly ash increased the volume of pores below 100 nm in the hardened slurry, mainly regarding the volume of the 0–20 nm pore group and 20–100 nm pore group.

#### 3.3.4. Thermogravimetric Analysis

Thermogravimetric analysis is one of the most commonly used methods to analyze the hydration mechanism of cement [28]. This method allows both the qualitative analysis of the hydration products of cement and the quantitative analysis of Ca(OH)_2_, the main hydration product of cement. The level of Ca(OH)_2_ content in the system affects the stability of the hydration products, especially when the cement is replaced by mineral admixtures; thus, it is important to study the content of Ca(OH)_2_ in the hardened slurry. As can be seen from the figure, there are three main heat absorption peaks on the DTG curve for pure cement specimens in the temperature range below 900 °C: the C-S-H gel and calcium alumina dehydration stage (50–200 °C), the Ca(OH)_2_ dehydration stage (400–550 °C), the CaCO_3_ dehydration stage (600–800 °C), and the main source of CaCO_3_ in the CaCO_3_ dehydration stage (600–800 °C). This is the product of carbonation occurring in cement, and Ca(OH)_2_ reacts with CO_2_ in air to form CaCO_3_. The first two peaks represent weight loss, while the third peak represents weight absorption.

Figure 14 shows the DTG curve and Ca(OH)_2_ content of the hardened slurry of composite cementitious material containing fly ash at a water to glue ratio of 0.45 and a curing time of 28 d, under standard curing conditions. According to Figure 14, it can be seen that the thermogravimetric curve of the hardened slurry containing fly ash is consistent with the variation pattern of the pure cement specimen, and there are mainly three main heat absorption peaks. The heat absorption peak of Ca(OH)_2_ on the DTG curve becomes weaker with the increase in fly ash admixture. This indicates that Ca(OH)_2_ is mainly produced by cement hydration and consumed by the volcanic ash reaction of fly ash at the later stage of hydration. The heat absorption peaks of C-S-H gel and calcium alumina on the DTG curve become weaker with the increase in fly ash admixture, and the higher the admixture amount, the weaker the heat absorption peaks. In particular, the peak degree of C-S-H gel and calcium alumina on specimens F20, F30, and F40 becomes significantly smaller. This indicates that at 10% fly ash admixture, fly ash can effectively promote the hydration reaction of cement and compensate for the decrease in the amount of hydration products produced partially by the loss of cement clinker.

From the figure, it can be found that the incorporation of fly ash will directly affect the C-S-H gel, calcium alumina, and Ca(OH)_2_ components, which will cause a decrease in the products of the hydration reaction due to the replacement of cement clinker components, but the volcanic ash effect of consuming Ca(OH)_2_ to generate new hydration products will occur with fly ash, which is mainly reflected in the increase in strength at a later stage. Moreover, from the third heat absorption peak, it can be seen that a small amount of fly ash admixture is beneficial for cement hydration and has a certain promotion effect.

## 4. Conclusions

In order to study the physical and chemical properties of cement-based grouts mixed with different amounts of fly ash, we conducted a comprehensive study in terms of four aspects, namely rheological properties, strength properties, hydration properties, and microscopic mechanisms, and the main findings are as follows.

The inclusion of fly ash reduces the thixotropic area of the composite cement slurry, which is beneficial to the pumping in the pipeline conveying process. The inclusion of fly ash reduces the yield stress and plastic viscosity of the cement slurry, but the rheological index increases and then decreases with the increase in fly ash, and the composite slurry has the lowest degree of shear thinning with 30% fly ash admixture.The incorporation of fly ash reduced the hydration exothermic rate and total hydration exothermic amount of the composite slurry and prolonged the hydration induction period, but the promotion effect of fly ash on the hydration rate of cement was obvious with 10% fly ash admixture. The admixture of fly ash increased the empty volume of the composite slurry, but it had little effect on the most probable aperture, and the porosity of the system increased, which led to a decrease in compressive strength.The effect of adding fly ash on hydration products is mainly reflected by the C-S-H gel produced by cement hydration and the change in calcium alumina and Ca(OH)_2_. Fly ash is not directly involved in the hydration reaction of cement, but it can promote cement hydration and increase the reaction rate of cement.By analyzing the rheological properties, mechanical properties, and hydration properties of fly ash composite cement slurry, a comprehensive analysis was performed that found that the rheological properties are better when fly ash is mixed with 20–30%, and the water to glue ratio can be reduced to improve the strength without affecting the pumping demand.

## Figures and Tables

**Figure 1 materials-15-08804-f001:**
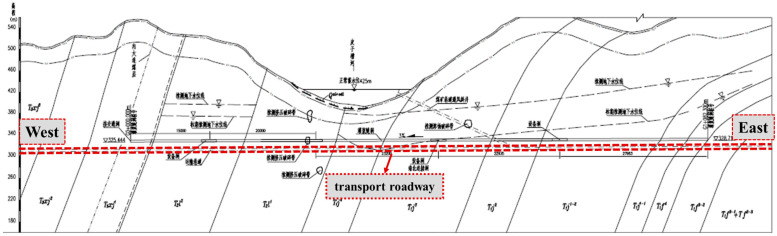
Longtan coal mine transportation roadway grouting and seepage control treatment area profile.

**Figure 2 materials-15-08804-f002:**
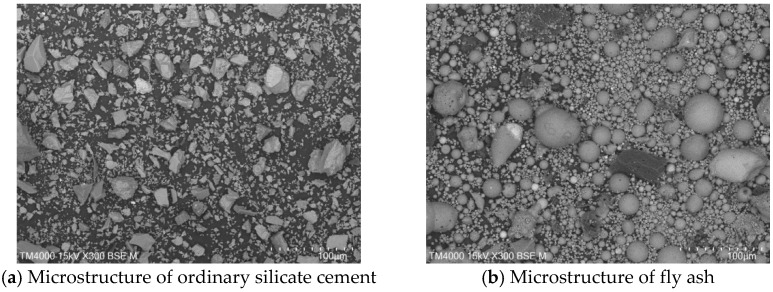
Raw material microscopic morphology.

**Figure 3 materials-15-08804-f003:**
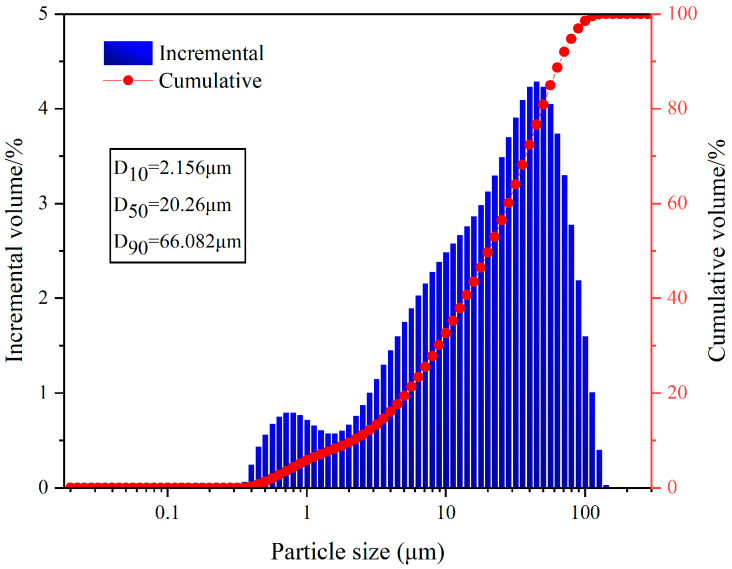
Particle size distribution of fly ash.

**Figure 4 materials-15-08804-f004:**
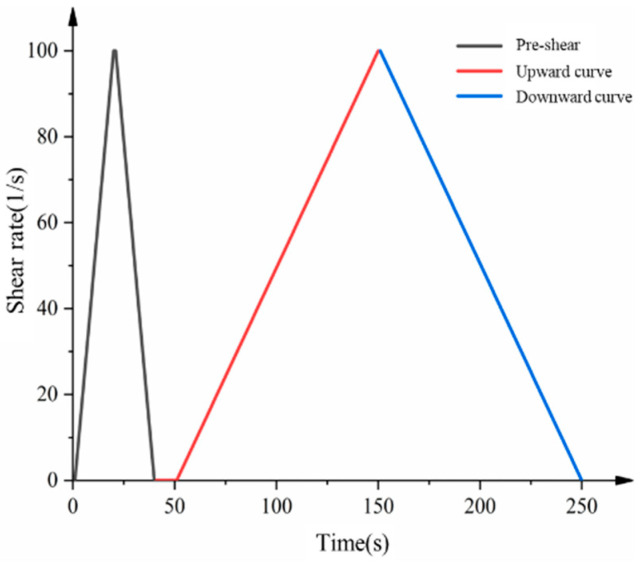
Rheological test procedure.

**Figure 5 materials-15-08804-f005:**
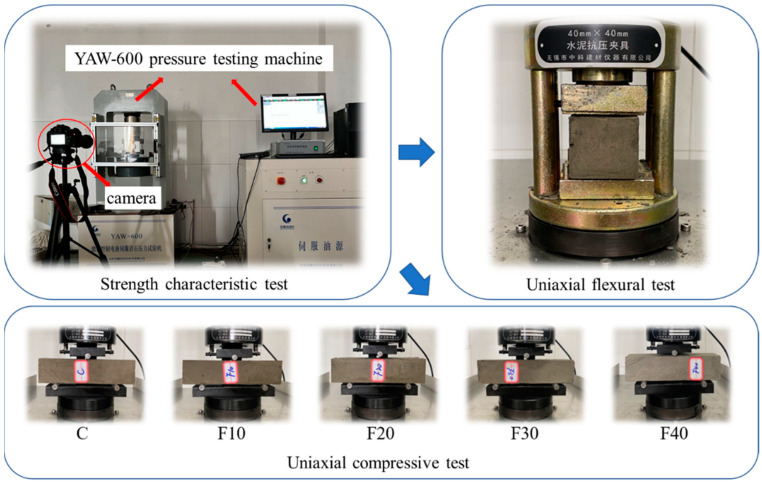
Strength characteristics test.

**Figure 6 materials-15-08804-f006:**
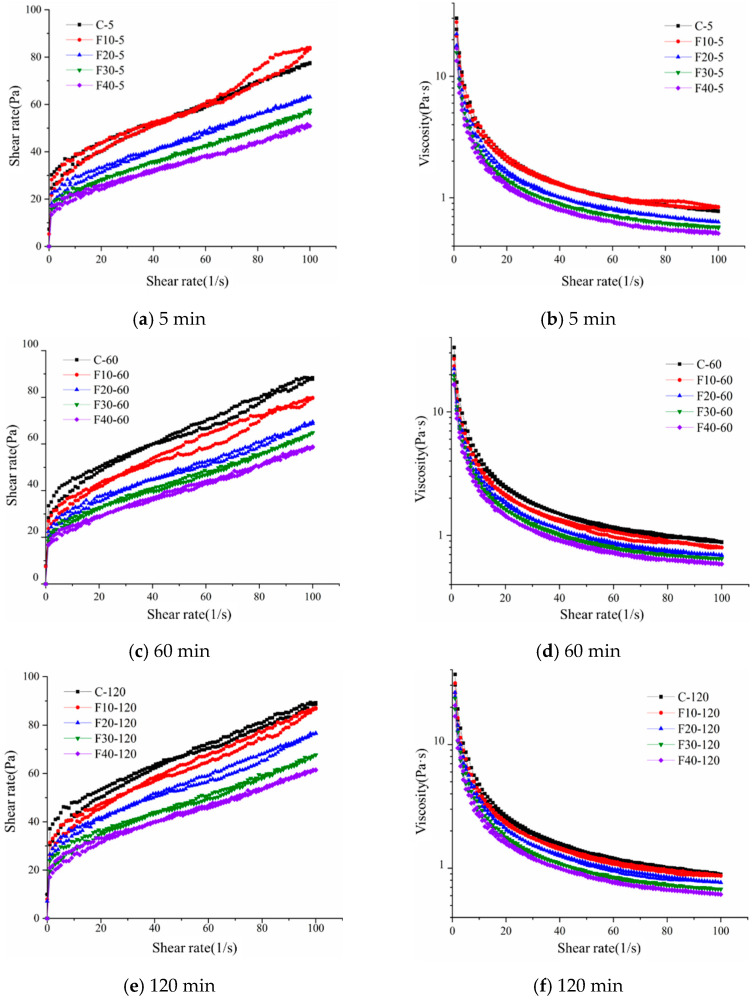
Thixotropic ring curve of composite cement slurry containing fly ash.

**Figure 7 materials-15-08804-f007:**
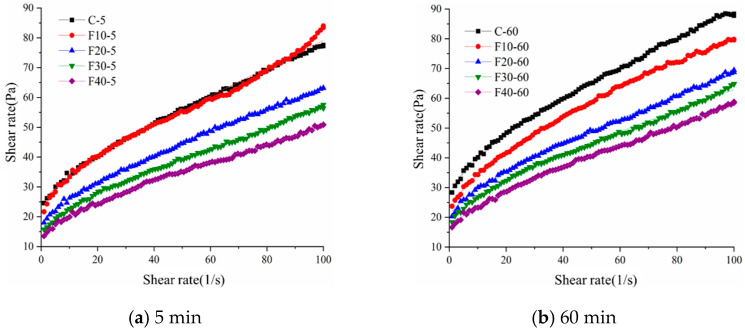
Rheology curve of composite cement slurry containing fly ash.

**Figure 8 materials-15-08804-f008:**
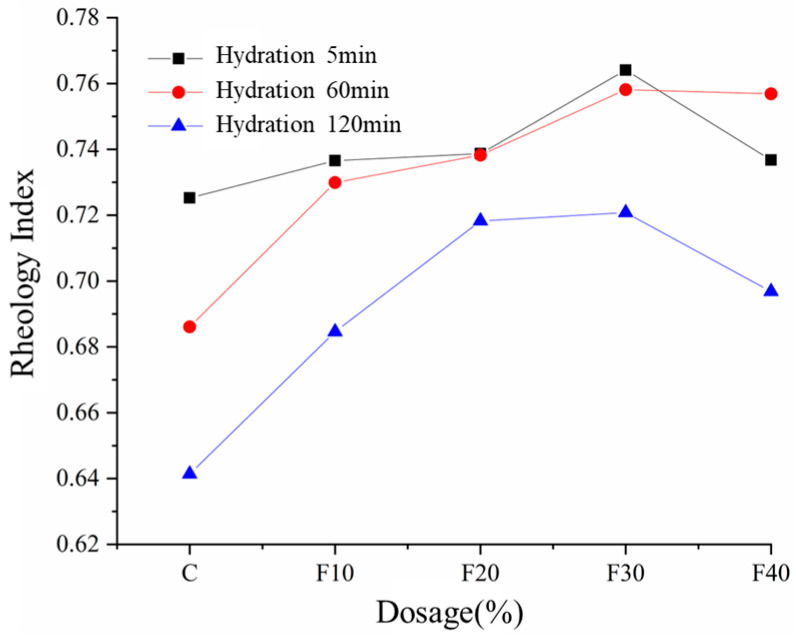
Rheological index of composite cement slurry.

**Figure 9 materials-15-08804-f009:**
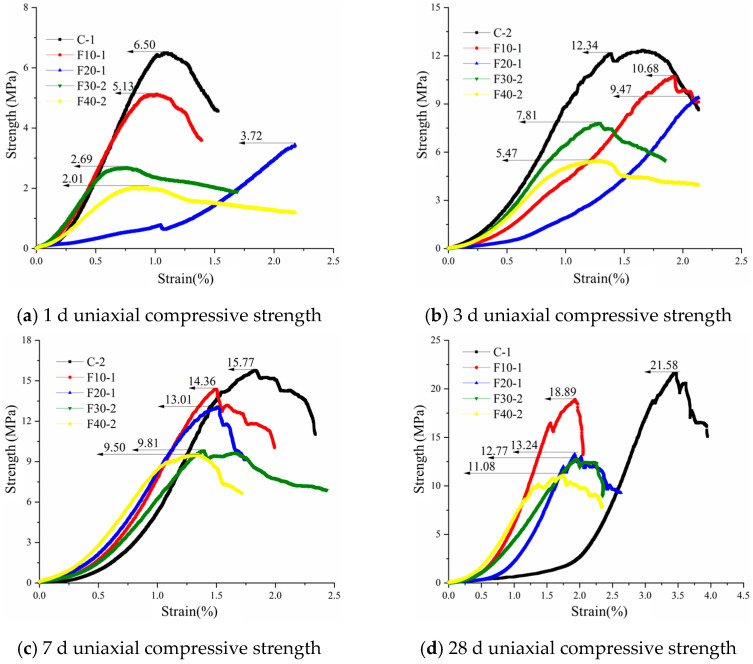
Compressive strength of cement-based grouting materials containing fly ash at different ages.

**Figure 10 materials-15-08804-f010:**
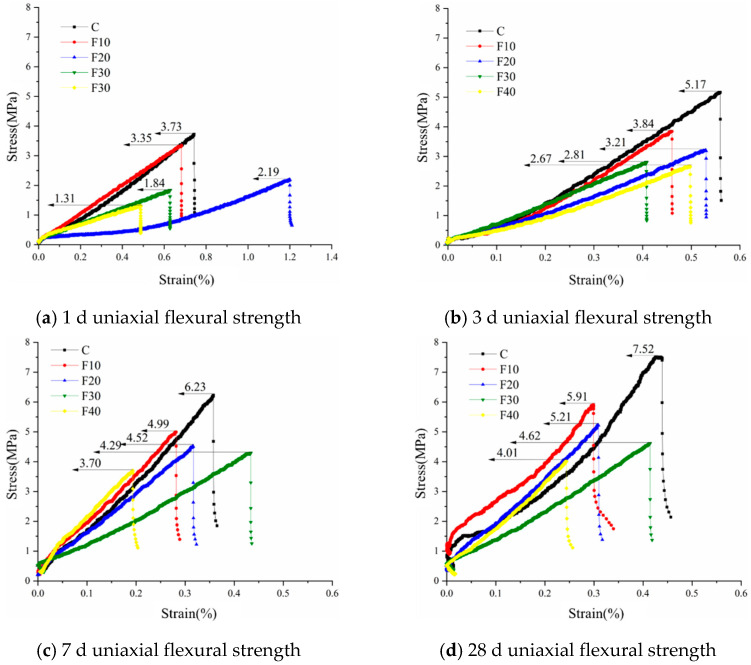
Flexural strength of cement-based grouting materials containing fly ash at different ages.

**Figure 11 materials-15-08804-f011:**
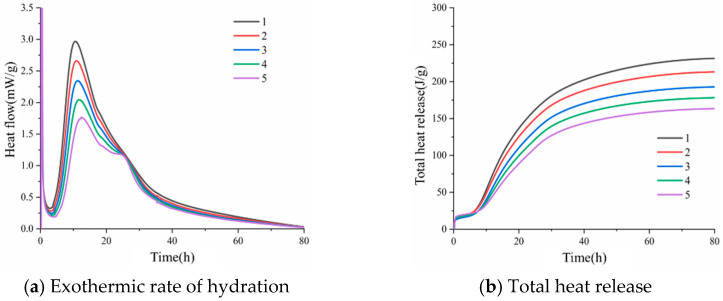
Heat of hydration of composite cementitious materials containing fly ash.

**Figure 12 materials-15-08804-f012:**
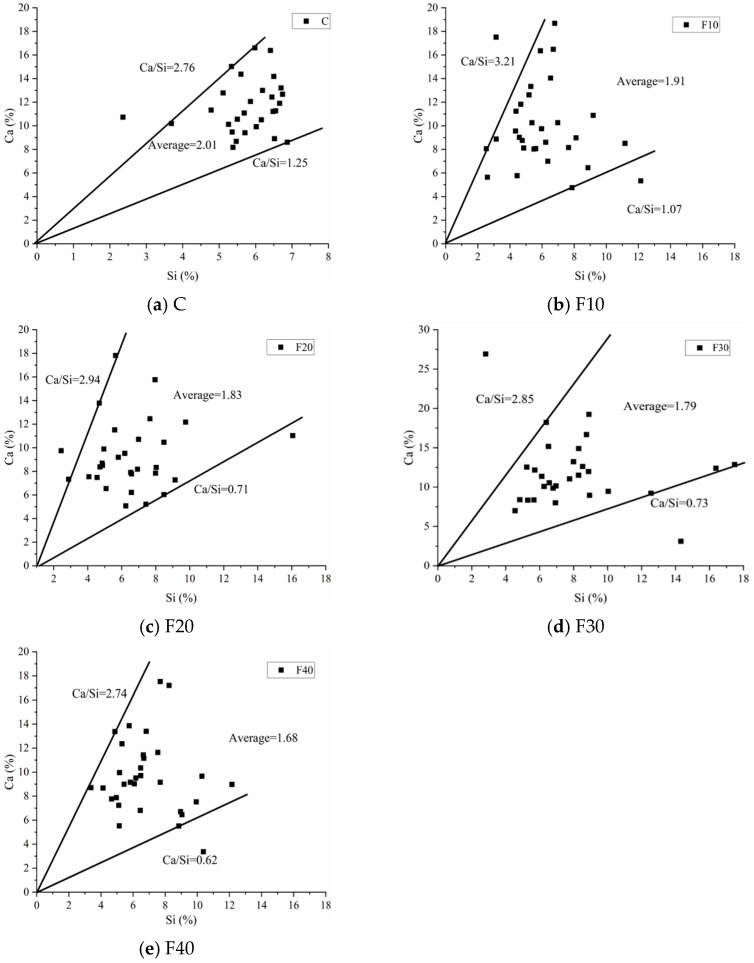
Ca/Si of the hardened slurry at the age of 3 d.

**Figure 13 materials-15-08804-f013:**
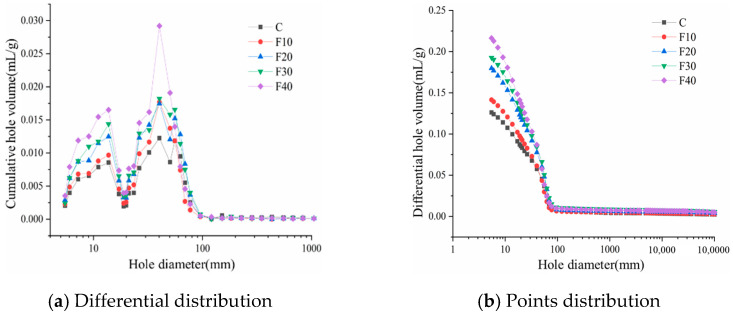
28 d pore structure of hardened slurry of composite cementitious material containing fly ash.

**Figure 14 materials-15-08804-f014:**
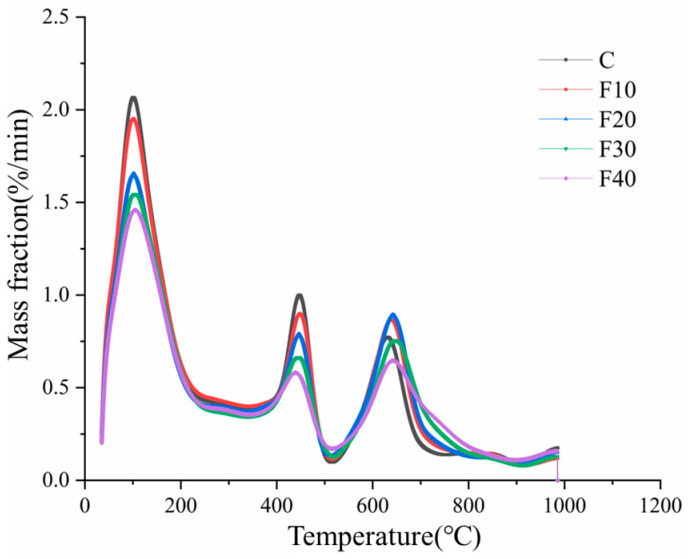
DTG curve of fly ash composite cementitious material-hardened slurry with water to glue ratio 0.45, maintenance time 28 d.

**Table 1 materials-15-08804-t001:** Cement physical and mechanical properties.

Standard Consistency of Water/%	Initial Setting Time/min	Final Setting Time/min	Density/(g/cm^2^)	Flexural Strength/MPa		Compressive Strength/MPa		Specific Surface Area/(m^2^/kg)
				**3**	**28**	**3**	**28**	
28.4	185	240	3.03	6.5	9.0	29.5	58	365

**Table 2 materials-15-08804-t002:** Cement chemical composition.

Grouping	CaO	SiO_2_	Al_2_O_3_	Fe_2_O_3_	MgO	SO_3_
Percentage/wt%	51.42	24.99	8.26	4.03	3.71	2.51

**Table 3 materials-15-08804-t003:** Fly ash chemical composition.

Grouping	SiO_2_	Al_2_O_3_	Fe_2_O_3_	CaO	K_2_O	TiO_2_	MgO	Na_2_O	SO_3_	P_2_O_5_	Cl	NiO
Percentage/wt%	53.97	31.15	4.16	4.01	2.03	1.13	1.01	0.88	0.73	0.67	0.13	0.11

**Table 4 materials-15-08804-t004:** Cement–fly ash net slurry ratio.

Specimen	Composition of Cementitious Materials (Quality Percentage/wt%)		Water to Glue Ratio
	Pure Cement	Fly Ash	
C	100	0	0.45
F10	90	10	
F20	80	20	
F30	70	30	
F40	60	40	

**Table 5 materials-15-08804-t005:** Rheological parameters of composite slurry containing fly ash.

Specimen	Hydration Time/min					
	5		60		120	
	*τ*_0_/Pa	*μ*/(pa·s)	*τ*_0_/Pa	*μ*/(pa·s)	*τ*_0_/Pa	*μ*/(pa·s)
C	23.50388	0.59145396	27.12297	0.690014053	28.19716	0.696636506
F10	22.8594	0.576337484	20.60124	0.67780415	25.37972	0.680796783
F20	17.38168	0.497821347	20.43129	0.522118643	24.24793	0.570086491
F30	15.40472	0.443064981	18.86049	0.479128569	19.29076	0.516306447
F40	13.22972	0.401255591	16.20314	0.450013833	16.73296	0.486514278

**Table 6 materials-15-08804-t006:** Basic characteristics of hydration products.

Name	Relative Density	Degree of Crystallization	Morphology	Size
C-S-H	2.3–2.6	Poor	Fibrous, network-like, etc., not easy to distinguish in late hydration	1 μm × 0.1 μm, thickness less than 0.01 μm
Ca(OH)_2_	2.24	Fine	Striped	0.01–0.1 mm
Calcareous alumina	1.75	Good	Needle stick	10 μm × 0.5 μm
Monosulfur-type hydrated calcium sulfur aluminate	1.95	General	Hexagonal lamellar, irregular petal shape	1 μm × 1 μm × 0.1 μm
C-S-H	2.3–2.6	Poor	Fibrous, network-like, etc., not easy to distinguish in late hydration	1 μm × 0.1 μm, thickness less than 0.01 μm

**Table 7 materials-15-08804-t007:** Microscopic morphology of cement–fly ash composite cementitious material-hardened slurry at the age of 3 d under standard curing conditions.

Specimen	Microscopic Morphology of Hardened Slurry at the Age of 3 d under Different Magnifications	
	500×	1500×
C	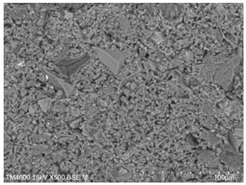	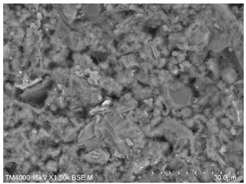
F10	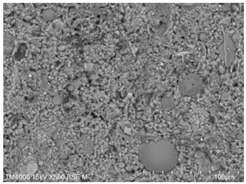	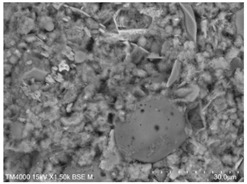
F20	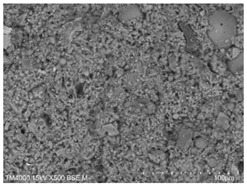	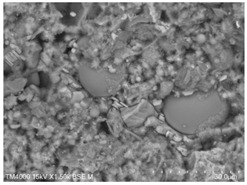
F30	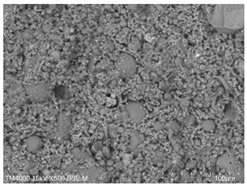	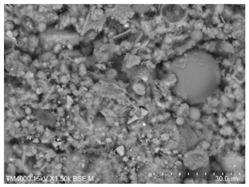
F40	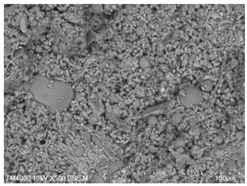	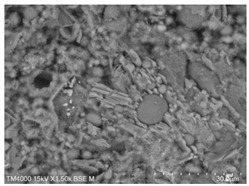

**Table 8 materials-15-08804-t008:** Microscopic morphology of cement–fly ash composite cementitious material-hardened slurry at the age of 7 d under standard curing conditions.

Specimen	Microscopic Morphology of Hardened Slurry at the Age of 7 d under Different Magnifications	
	500×	1500×
C	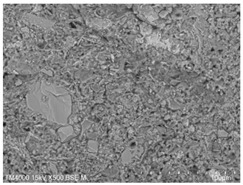	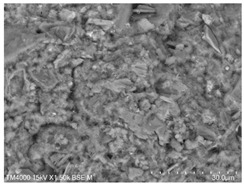
F10	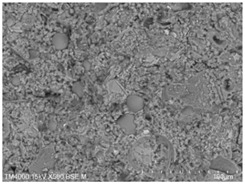	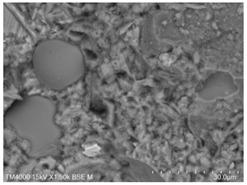
F20	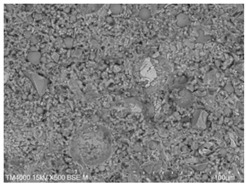	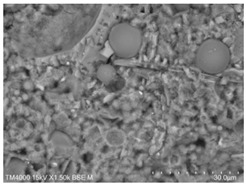
F30	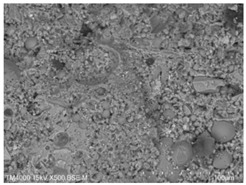	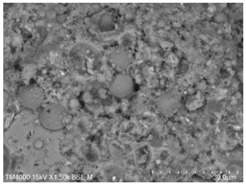
F40	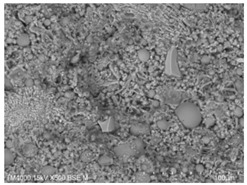	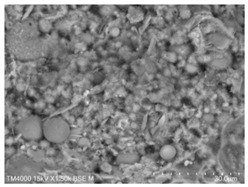

## Data Availability

Not applicable.

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
