# Peer review of "Study on the Physical and Chemical Properties of Cement-Based Grout Containing Coal–Fly Ash"

_materials, 2022, doi:10.3390/ma15248804_

Round 1
Reviewer 1 Report
The paper is interesting, well written and the results are appropriately explained. Nevrtheless, the text needs a few corrections and integrations.
-The title should be changed in:
Study on the physical and chemical properties of cement-based grout containing coal-fly ash
-There are other types of fly-ash that can be recycled. An example is: DOI 10.1089/109287501750281059. The author could report in the references as an example.
-The origin of fly-as should be reported briefly in the introduction
-page 3: the authors should write that glue is the mix cement/fly ash (if it is)
-page 3: which is the meaning of secondary fly ash? Is there a primary fly-ash?
-Table 1: is it quite strange that the water to glue ratio is the same for all the mixes. Generally, water is added until the mix reaches a good workability, subjectively evaluated by the operator
-Equation, page 4: m is a constant? In that case, the last squared method applied to the experimental data should result in a numerical value.
-Page 6: the authors should explain what a falling curve is for the less expert readers
Page 9-10: the difference between a thixotropic ring curve and a rheological curve should be explained, because Figure 6 could be omitted, as Figure 5 can give all the information needed. It is not clear to me which is the difference between Figure 5 (c) and Figure 6 (b). The pure cement slurry is more pseudoplastic than when it is added to fly ash. This is demonstrated by the lowest Rheology Index.
-Figure 7: a figure should be self-consistent. The authors should write inside the figure the meaning of 5-60-120
-page 11: it is obvious that the compressive strength of a cementitious mix increases with the age. Hence the sentence: “It can be seen from……..show this pattern” could be omitted
Figure 13: the first two peaks should represent a loss of weight while the third peak should be an uptake of weight. The authors should report that in the text.
Reviewer 2 Report
The paper study physical and chemical properties of cement-based grout containing fly ash.
I suggest to authors to make many corrections and rewrite the manuscript in less confusing form with higher research level, such as:
1. Abstract - "fly ash, secondary fly ash" - Why this repetition?
- the abstract is a very chaotic mixture of statements - it is necessary to rewrite the text of the abstract into coherent sentences containing the main results and benefits.
2. Introduction - there is many irregularities, such as "Fly ash is made by ultra-fine grinding process" - "Fly ash is a fine powder that is a byproduct of burning pulverized coal in electric generation power plants."
- What means expression "Secondary fly ash?"
3. Materials and Methods: "production scale of 1.5 million" - of what?
- "2.1.1. Cement" - you should state the basic properties of cement, everyone knows that the cement is consist of C3S, C2S, C3A, C4AF.
- " Figure 2a" is it cement used in the research?
- It is neccessary to add particle size distribution and chemical composition of the fly ash.
- What does "Water to glue ratio" mean?
- Tabel 1: "/%)" - please correct
- Figure 3 - enlarge the image to make it readable
- Were "Uniaxial compressive and uniaxial flexural strength tests" provides according any standard?
- page 7 - "there are a lot of unnecessary sentences, such as The thixotropic ring curve consists of a rising curve and a falling curve"
- Figure 8 - It should be Strength (MPa) in axis y
- Discussion of results is totally missing. Kindly add proper references for each experimental discussion.
- Why the axis y in Figure 13 is in "Mass fraction (%C)? It is very confusing...
4. Conclusion - point 3 is not true....fly ash, generally, participate in hydration reactions - you should study Pozzolanic reaction!!
Reviewer 3 Report
This work is interesting to see the secondary fly ash was mixed with silicate cement to make grout in order to study the physical and chemical properties of grout, to select the grout with reasonable proportion, and to reduce the cement dosage. Besides, this is also a great research significance and economic value since authors mentioned the background of this study relies on the Longtan coal mine transport lane. It showed that different ratio of secondary fly ash affect the rheological properties, strength properties, hydration properties, and etc of the composite slurry. However, there are some issues that should be addressed before publication.
1. In Page 3, Section 2.1 Raw Materials. It is suggested to put the chemical composition of the cement and fly ash that used in this study.
2. In Page 10, Section 3.1.2. Rheological Parameters. Please explain why lower value of yield stress (20.60124 Pa) of 60 hydration time for F10 is obtained in comparison with 5 hydration time yield stress (22.8594 Pa) for the same composition?
2. In Page 14, Section 3.3.2 Microscopic morphology. Please indicate the “The bonding between the hydration products is not tight and pores can be clearly observed” from the micrograph.
Round 2
Reviewer 2 Report
Authors incorporated all of the comments in the paper.